# Quantum squeezing amplification with a weak Kerr nonlinear oscillator

Yanyan Cai[1,2,4], Xiaowei Deng[2,4], Libo Zhang[1,2,4], Zhongchu Ni[2], Jiasheng Mai[1,2], Peihao Huang[2], Pan Zheng[2], Ling Hu[2,3], Song Liu[2,3], Yuan Xu[2,3] ✉ & Dapeng Yu[2,3]

Quantum squeezed states, with biased quantum noise, have been widely utilized in quantum sensing and quantum error correction applications. However, generating and manipulating these nonclassical states with a large squeezing degree typically requires strong nonlinearity, which inevitably induces additional decoherence that diminishes the overall performance. Here, we demonstrate the generation and amplification of squeezed states in a superconducting microwave cavity with weak Kerr nonlinearity. By subtly engineering an off-resonant microwave drive, we observe cyclic dynamics of the quantum squeezing evolution in a displaced frame of the cavity. Furthermore, we deterministically realize quantum squeezing amplification by alternately displacing the Kerr oscillator using the Trotterization technique, achieving a maximum squeezing degree of 14.6 dB and a squeezing rate of 0.28 MHz. Our demonstrated displacement-enhanced squeezing operation offers a hardware-efficient approach for generating large squeezed states, promising potential applications in quantum-enhanced sensing and quantum information processing.

The landscape of quantum information science has been substantially enriched by the utilization of nonclassical bosonic states, which are essential for continuous variable quantum information processing[1,2]. Among these states, squeezed states[3], which exhibit reduced noise in one quadrature but increased noise in the conjugate one, stand out for their remarkable ability to enhance the measurement sensitivity beyond the standard quantum limit[4,5]. This enhancement is particularly critical in sensing applications[6,7], such as the search for elusive dark matter particles[8] and the observation of gravitational waves[9,10]. In addition, these nonclassical squeezed states serve as essential resources to generate Gottesman-Kitaev-Preskill states[11,12], squeezed cat states[13,14], and squeezed Fock states[15] for quantum error correction protocols[16–19].

Over the past few decades, the generation and manipulation of squeezed states have been successfully demonstrated on a variety of physical platforms[20], including optical and microwave photons[21–28], as well as mechanical and acoustic phonons[29–33]. The underlying

mechanism for generating these nonclassical states generally involves engineering a nonlinear interaction within these bosons. The pursuit of a large squeezing typically necessitates a strong nonlinearity, which, however, inevitably introduces additional decoherence and limits the accessibility of the large Hilbert space of an oscillator with a linear displacement drive.

In the realm of quantum optics with superconducting microwave circuits[34], Kerr nonlinearity can be achieved using Josephson junctions, which are essential elements for developing parametric amplifiers[35,36], generating itinerant squeezed microwave fields[21,27], and realizing superconducting qubits[37]. By coupling to a strong nonlinear superconducting qubit, a harmonic oscillator in the single-photon Kerr regime has been demonstrated to realize the collapse and revival of coherent states[38] and to autonomously generate microwave Fock states[39,40]. In addition, the use of weak Kerr nonlinearities has recently been proposed for manipulating photon-blockaded states[41,42], with the operation speed enhanced by a large displacement. A similar idea was

---

[1]Southern University of Science and Technology, Shenzhen, China. [2]International Quantum Academy, Shenzhen, China. [3]Shenzhen Branch, Hefei National Laboratory, Shenzhen, China. [4]These authors contributed equally: Yanyan Cai, Xiaowei Deng, Libo Zhang. ✉e-mail: xuyuan@iqasz.cn

also demonstrated to implement fast conditional operations[26,43]. However, despite these theoretical and experimental advancements, demonstrating the fast generation of large squeezed states with a weak Kerr nonlinear oscillator remains a formidable experimental challenge.

In this work, we experimentally demonstrate the generation and amplification of squeezed states in a superconducting microwave cavity with weak Kerr nonlinearity. By subtly engineering an off-resonant microwave drive on the Kerr oscillator, we experimentally observe the cyclic dynamics of the squeezing evolution in a displaced frame without state collapse during the evolution. In addition, we employ the Trotterization technique to realize quantum squeezing amplification by alternately displacing the Kerr oscillator, achieving a maximum squeezing degree of 14.6 dB, which, to our knowledge, is the largest squeezing value for microwave photonic states within the cavity. This technique offers an efficient approach for realizing displacement-enhanced squeezing operations to generate large squeezed states. Furthermore, we demonstrate the metrological advantages of these nonclassical states for sensing small displacements, promising potential applications in quantum-enhanced metrology.

## Results

A linear harmonic oscillator has a quadratic potential well, which can be quantized into discrete and equally spaced energy levels, as depicted in Fig. 1a. The oscillator would induce a self-Kerr nonlinearity when coupling to a nonlinear superconducting qubit in circuit quantum electrodynamics[37]. This nonlinear Kerr oscillator has slightly

unequally spaced energy levels with the Hamiltonian expressed as $H_K = -\frac{K}{2} a^{\dagger 2} a^2$ (assuming $\hbar = 1$) in the rotating frame of the oscillator frequency $\omega_c$. Here, $a$ ($a^\dagger$) is the annihilation (creation) operator, and $K$ is the strength of the Kerr nonlinearity of the oscillator. Under this Kerr Hamiltonian, a coherent state will first evolve into a slightly crooked squeezed state and then into a complete phase collapse state[38], as indicated in Fig. 1b.

To avoid the state collapse and further improve the squeezing degree, we engineer an off-resonant microwave drive on the Kerr oscillator with a drive frequency of $\omega_d$, resulting in a driven Kerr oscillator

$$H_d = \Delta_d a^\dagger a - \frac{K}{2} a^{\dagger 2} a^2 + \Omega_d (a + a^\dagger), \qquad (1)$$

where $\Delta_d = \omega_c - \omega_d$ represents the frequency detuning between the oscillator and the drive, and $\Omega_d$ is the strength of the drive (ignoring the drive phase for simplicity without any loss of generality). This Hamiltonian represents the well-known prototypical model of a quantum Duffing oscillator, exhibiting nonequilibrium phase transition dynamics[44] and nonclassical Wigner negativity[45].

The quantum dynamics of the driven Kerr Hamiltonian with engineered parameters can be employed to generate two-photon squeezing (Fig. 1c), which can be further amplified through the Trotterization technique (Fig. 1d). To intuitively understand the squeezing

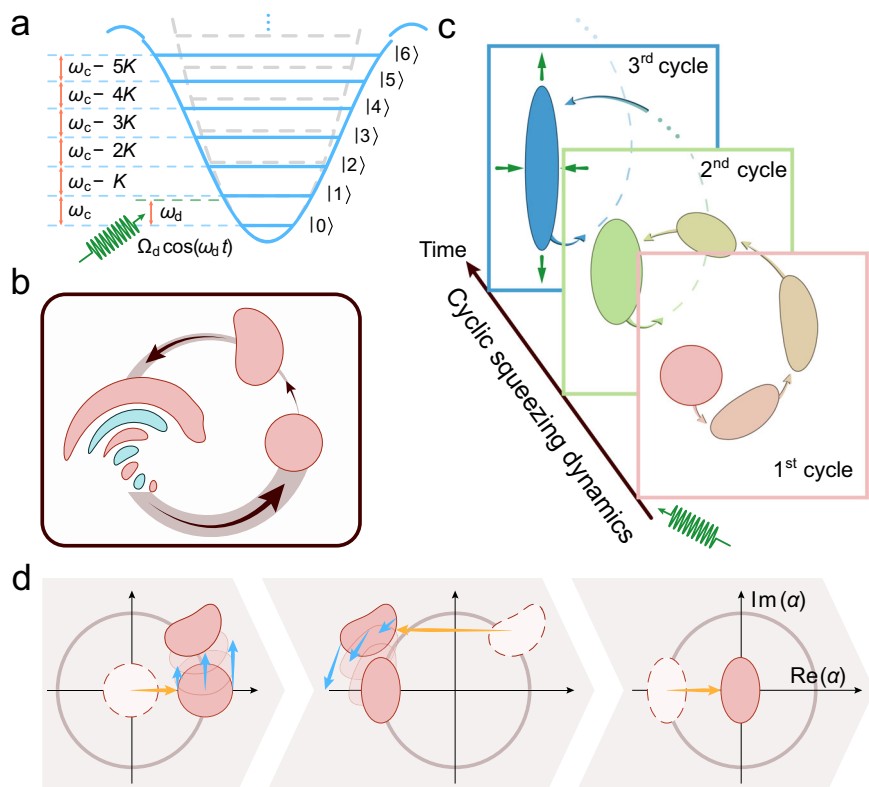

**Fig. 1 | Schematic illustration for achieving quantum squeezing amplification with a Kerr nonlinear oscillator. a** Energy level diagram of a linear harmonic oscillator (gray dashed lines) and a Kerr nonlinear oscillator (solid blue lines) with an engineered detuned drive of frequency $\omega_d$ and amplitude $\Omega_d$ (green wavy line). **b** The quantum evolution of a coherent state represented with Wigner functions in phase space, exhibiting squeezing and collapse without any drives on the Kerr oscillator. **c** The quantum evolution of a coherent state to realize cyclic squeezing amplification with an engineered detuned drive on the Kerr oscillator. The coherent state undergoes a cyclic evolution in the phase space and becomes a squeezed state at the final moment of each cycle, where the squeezing degree gradually increases with the number of cycles. **d** The quantum evolution in phase space with the Trotterization technique using displacement operations (yellow arrows) and Kerr-nonlinear rotations (blue arrows), showing the transition from a coherent state to a squeezed state.

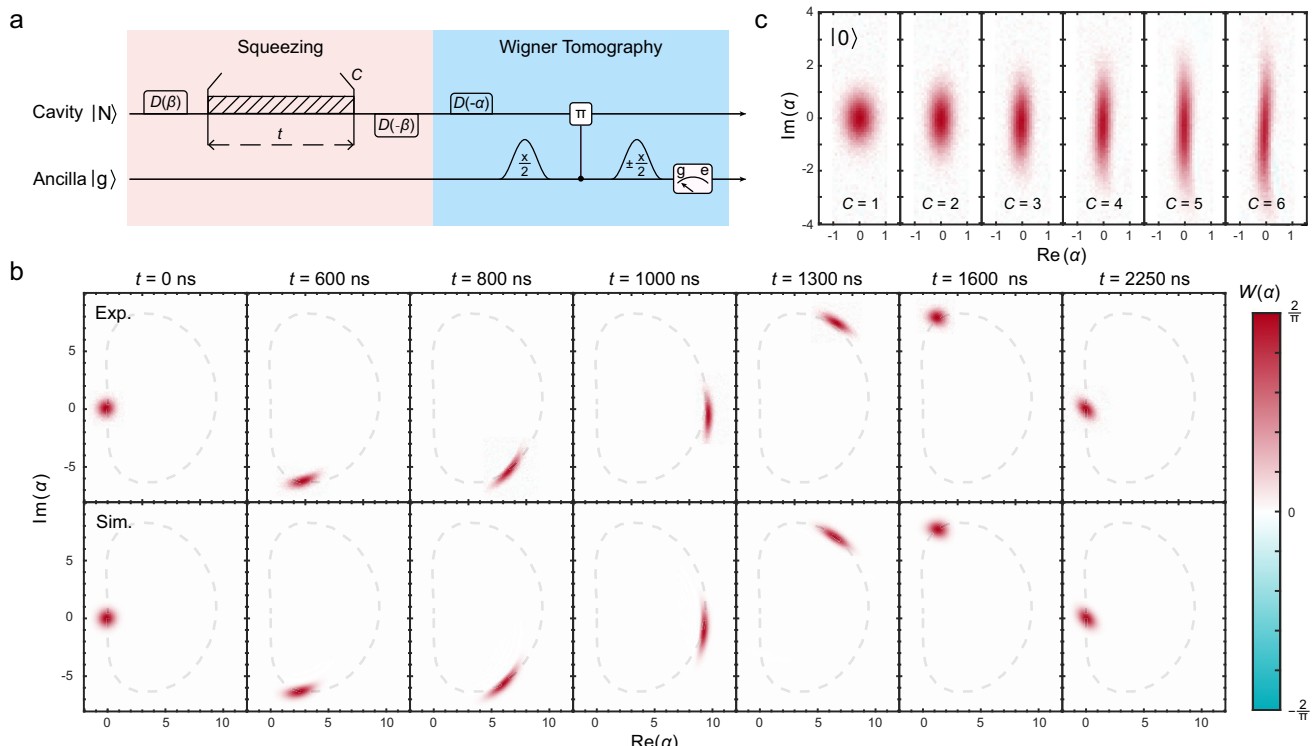

**Fig. 2 | Cyclic squeezing evolution with a detuned drive on the Kerr nonlinear oscillator for generating squeezed vacuum states. a** Experimental sequence for cyclic squeezing evolution and characterization with Wigner tomography. **b** Experimentally measured (top row) and numerically simulated (bottom row) Wigner function snapshots for the cyclic squeezing evolution of a vacuum state

$|N = 0\rangle$ in the first cycle (C = 1), showcasing that the quantum state no longer collapses during the evolution and evolves into a squeezed state at the final moment of each cycle. **c** Experimentally measured Wigner functions of the generated squeezed vacuum states at the final moment of the first 6 evolutionary cycles, obviously indicating that the squeezing is enhanced as the number of cycles increases.

mechanism, we move into a displaced frame with a unitary operation of $D(\beta) = \exp(\beta a^\dagger - \beta^* a)$. Then, the aforementioned driven Kerr Hamiltonian can be transformed into

$$H_\beta = \Delta' a^\dagger a - \frac{K}{2} a^{\dagger 2} a^2 - \frac{K}{2}(\beta^2 a^{\dagger 2} + \beta^{*2} a^2) \\ - K\beta a^\dagger (a^\dagger a - r) + \text{H.C.} \qquad (2)$$

with $\Delta' = \Delta_d - 2K|\beta|^2$ and $r = \Delta_d/K + \Omega_d/K\beta - |\beta|^2$ representing the photon-blockade parameter. The first line of the above equation corresponds to the Hamiltonian of a Kerr parametric oscillator, which promises potential applications in quantum computation[46–49], quantum metrology[50], quantum phase transition[51,52], and quantum tunneling[53,54].

## Cyclic squeezing dynamics

Here, we first discuss the driven squeezing dynamics. By properly engineering the detuned drive on the Kerr oscillator, we find that a coherent state no longer collapses during the evolution and undergoes a cyclic squeezing evolution, as shown in Fig. 1c. Intuitively, the photon blockade term in the displaced frame arises from the interplay between the detuned drive and the Kerr nonlinearity. The detuned drive acts as a driving force pulling the coherent state along the negative imaginary axis in phase space, while the Kerr term rotates the state anticlockwise at a rate proportional to the average photon number. Carefully balancing these two effects prevents state collapse and enables periodic cycling trajectory dynamics (see Supplementary Fig. 2). The mirror-symmetry trajectory around the real axis in phase space effectively cancels the photon blockade effect at the end of each cycle, achieving the displacement-enhanced squeezing operation with a squeezing rate $K\beta^2$, as indicated by the two-photon squeezing term in

Eq. (2). The optimal parameters of the detuned drive are extracted from numerical simulations by evaluating the fidelity between the evolved squeezed state and an ideal squeezed state $|\xi\rangle = S(\xi)|0\rangle$, where $S(\xi) = \exp(\frac{\xi^*}{2} a^2 - \frac{\xi}{2} a^{\dagger 2})$ represents a squeezing operator. Here, $\xi$ is a complex number of the squeezing parameter and $20\log_{10}(e^{|\xi|})$ defines the squeezing level or squeezing degree in dB. The details of the numerical simulations are presented in Supplementary Note 2.

In our experiment, we realize a Kerr nonlinear oscillator by dispersively coupling a high-quality superconducting microwave cavity (resonance frequency of $\omega_c/2\pi = 6.60$ GHz) to an ancillary superconducting qubit (transition frequency of $\omega_q/2\pi = 5.28$ GHz). The cavity mode has a weak Kerr nonlinearity of $K/2\pi = 5.83$ kHz ($K/\omega_c < 10^{-6}$) and a single-photon lifetime of 395 μs (corresponding to a decay rate $\kappa_c/2\pi = 0.40$ kHz), serving as the storage cavity for storing and manipulating the multiphoton squeezed states. The ancillary qubit mode, with an energy relaxation time of about 38 μs and a Ramsey (echo) decoherence time of 50 (58) μs, is utilized for characterizing the generated squeezed states in the storage cavity. Detailed information regarding the device parameters and the experimental setup is provided in Supplementary Note 1. Note that, our displacement-enhanced squeezing method is also applicable even with an ultra-weak Kerr nonlinearity $K < \kappa_c$ (see Supplementary Fig. 6).

The experimental sequence is shown in Fig. 2a, where the initial cavity vacuum state is first transformed into a displaced frame using a displacement operation $D(\beta)$, then evolves under the detuned driven Kerr Hamiltonian, and finally reverts to the original frame via a reverse displacement operation $D(-\beta)$. To achieve a larger squeezing degree, we choose an optimal displacement amplitude $\beta = 2$, and drive frequency detuning $\Delta_d/2\pi = 56$ kHz and amplitude $\Omega_d/2\pi = 2.01$ MHz, as determined from numerical simulations (see Supplementary Fig. 3). At the end of the sequence, we employ Wigner tomography to

characterize the cavity states. During the first evolution cycle with a period of 2250 ns, we measure the Wigner functions of the cavity states at various moments, which are in good agreement with numerical simulations, as indicated in Fig. 2b. The measurement results reveal that the coherent state rotates without collapsing in phase space under the detuned driven Kerr Hamiltonian, and is compressed into an elliptical structure in phase space at the end of each evolution cycle, indicating the successful generation of squeezed states. We displace the compressed coherent state to the origin in phase space, perform a virtual phase rotation operation to eliminate the rotation angle (see Supplementary Note 4), and measure the Wigner functions of the generated squeezed states at the final moment of different cycles. The experimental results displayed in Fig. 2c, clearly show that as the number of cycles increases, the cavity state is gradually compressed in one direction and stretched in the orthogonal direction. This structural feature indicates that the squeezing level of the cavity state is progressively amplified with increasing cycles, demonstrating the effectiveness of the displacement-enhanced squeezing approach. In addition, the driven squeezing approach may offer potential applicability and flexibility in engineering complex squeezing evolution. For example, we employ this approach to generate squeezed multiphoton Fock states $S(\xi)|N\rangle$ with $N$ up to 6 (see Supplementary Fig. 7). The generated squeezed Fock states, with a large squeezing level and photon number, could potentially achieve higher sensitivities than either squeezed vacuum states or multiphoton-number Fock states alone, offering significant advantages in quantum-enhanced precision metrology.

## Squeezing amplification with Trotterization technique

To further enhance the squeezing levels without relying on brute-force optimizations, we propose utilizing the Trotterization technique[55] to eliminate the residual photon-blockade effect in Hamiltonian Eq. (2). This Trotterization strategy is based on the Kerr squeezing dynamics mentioned above, enabling a digital squeezing protocol that relies solely on simple, natively available gates and dynamics, as shown in Fig. 1d. The key point of this approach lies in the addition of the driven Kerr Hamiltonian in two opposite displacement frames, $H_\beta$ and $H_{-\beta}$, which could completely eliminate the photon-blockade term, resulting in a Kerr parametric oscillator Hamiltonian:

$$\begin{aligned} H_{\mathrm{KPO}} &= \frac{H_\beta + H_{-\beta}}{2} \\ &= \Delta' a^\dagger a - \frac{K}{2} a^{\dagger 2} a^2 - \frac{K}{2}\left(\beta^2 a^{\dagger 2} + \beta^{*2} a^2\right). \end{aligned} \tag{3}$$

Based on the Trotterization formula, the time evolution of the Kerr parametric oscillator Hamiltonian can be expressed as

$$e^{-iH_{\mathrm{KPO}}\delta t} = e^{-iH_{-\beta}\delta t/2} e^{-iH_\beta \delta t/2} + \mathcal{O}[(\delta t)^2], \tag{4}$$

in a discretized evolutionary time step $\delta t$ with Trotter errors suppressed to an order of $\mathcal{O}[(\delta t)^2]$. This Trotterization method alternates the phases of the displacement frame, which effectively cancels the undesired nonlinear photon-blockade phase errors while reinforcing the two-photon squeezing term. By increasing the displacement amplitude $\beta$, the two-photon squeezing term would dominate in the Kerr parametric oscillator Hamiltonian of Eq. (3) for achieving large squeezed states. Note that in this scenario, the rapid switching between opposite-signed displaced states prevents any single step of Kerr evolution from going far beyond the squeezing stages, rendering the detuned drive unnecessary during the Kerr evolution. We therefore set $\Omega_{\mathrm{d}} = 0$ to restrict the average photon numbers of the intermediate states and engineer a virtual phase shift after each evolution cycle to mitigate the Kerr-induced phase on the squeezed state (see Supplementary Note 4).

The experimental sequence to demonstrate the quantum squeezing amplification with the Trotterization method is shown in Fig. 3a. In this sequence, we alternately transform the Kerr oscillator into different displacement frames with opposite phases. This is similar to an echoed displaced Kerr evolution that would average away the undesired photon-blockade term and suppress cavity dephasing errors (see Supplementary Note 7). To evaluate the squeezing performance of the generated squeezed states, we measure one-dimensional (1D) Wigner function cuts along the real axis ($\mathrm{Im}(\alpha) = 0$) and the imaginary axis ($\mathrm{Re}(\alpha) = 0$), respectively, with the experimental results shown in Fig. 3b. As the Trotter steps increase, the linewidth of one trace is compressed below 1/2, while the other one is expanded beyond 1/2, indicating an effective squeezing of the quantum state. From a global fit to both traces, we extract the squeezing parameters $|\xi|$ of the generated squeezed states as a function of the number of Trotter steps for various displacement amplitudes $|\beta|$, with the experimental results shown in Fig. 3c.

The experimental results reveal that as the number of Trotter steps increases, the squeezing level is amplified towards a large squeezed state. The eventual saturation in squeezing levels is attributed to Kerr nonlinearity, verified by numerical simulations. By performing a linear fit to the experimental data within the linear region, we extract the squeezing rate as a function of $|\beta|^2$, as shown in the inset of Fig. 3c. As the displacement amplitude $|\beta|$ increases, the squeezing rate is enhanced accordingly, but deviates from the ideal relationship $K\beta^2$. This deviation is primarily due to the neglect of Kerr evolution during the displacement operation and the influence of higher-order Kerr nonlinearities. When these aspects are taken into account, the numerical simulation results align well with the experimental data.

Furthermore, we have measured the 2D Wigner functions of the generated states at various Trotter steps, as shown in Fig. 3d. The 2D Wigner fitting results demonstrate a maximum squeezing degree of 14.6 dB, which is larger than previous demonstrations of intracavity squeezing[24–26,28]. Despite the gradual twisting and stretching of the squeezed states in phase space due to Kerr nonlinearity, these states continue to exhibit large Fisher information (FI), which quantifies the maximum information that can be extracted about a parameter from the quantum state.

## Quantum metrology for sensing small displacement

To assess the practical utility of these generated squeezed states, we further conduct a quantum metrology experiment to measure small displacements along the position quadrature using these states. In this experiment, we employ a parity measurement to map the information about the displacement onto the qubit state, with the experimental sequence illustrated in Fig. 4a. From the measured qubit ground state population $P_g$, the classical FI is calculated with the formula $I_{\mathrm{F}}(\alpha) = \frac{1}{P_g(1-P_g)}\left(\frac{\mathrm{d}P_g}{\mathrm{d}\alpha}\right)^2$ and is bounded by quantum FI $I_{\mathrm{Q}} = 4e^{2|\xi|}$ for an ideal squeezed state $|\xi\rangle$. These FIs are explicitly linked to the variance of the estimated displacement amplitude via the Cramér-Rao inequality[56]

$$\Delta\alpha \geq \frac{1}{\sqrt{N_{\mathrm{meas}}I_{\mathrm{F}}}} \geq \frac{1}{\sqrt{N_{\mathrm{meas}}I_{\mathrm{Q}}}}, \tag{5}$$

where $N_{\mathrm{meas}}$ represents the number of independent measurements.

Using the parity measurement results from $N_{\mathrm{meas}}$ experiments, we estimate a sample of displacement amplitude $\alpha$ by leveraging the Bayesian inference (see Supplementary Note 5). The variance of the estimated displacement amplitude is calculated by the Allan deviation formula $\sigma_\alpha = \sqrt{\frac{1}{2(l-1)}\sum_{k=1}^{l-1}(\alpha_{k+1}-\alpha_k)^2}$ with $l = 10$ samples. In Fig. 4b, we plot the Allan deviation for displacement estimation versus $N_{\mathrm{meas}}$ using the squeezed states generated with different Trotter steps. In addition, Fig. 4c depicts the measured Allan deviation as a function of

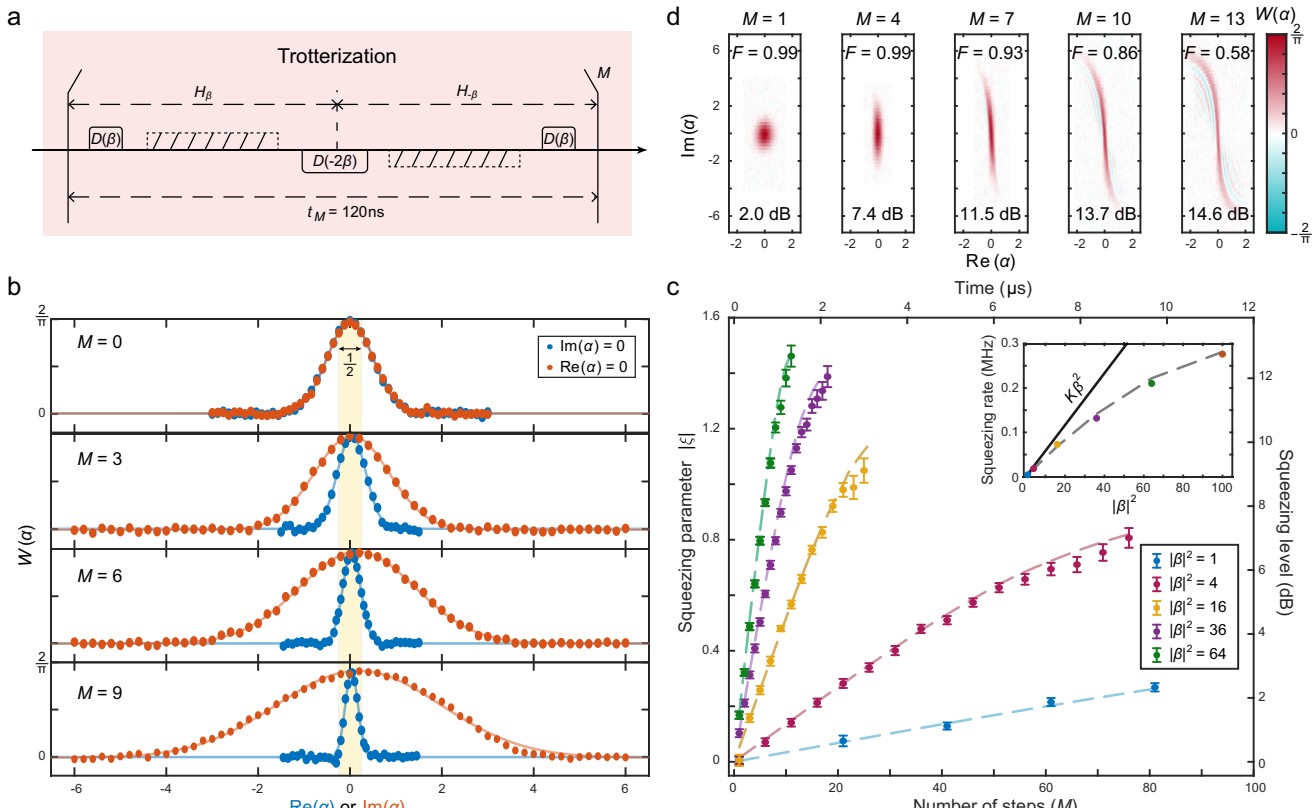

**Fig. 3 | Quantum squeezing amplification with the Trotterization technique.**
**a** Experimental sequence of the Trotter steps for realizing the squeezing amplification. **b** Measured 1D Wigner function cuts along the real axis (blue symbols) and the imaginary axis (red symbols) of the generated squeezed states for various Trotter steps. Solid lines are a global fit to these two traces at each Trotter step. **c** Experimentally extracted (symbols) and numerically simulated (dashed lines) squeezing parameters as a function of the number of Trotter steps (bottom axis) or

the evolutionary time (top axis) for various displacement amplitudes $|\beta|^2$. Error bars are the estimated 95% confidence intervals of the fittings. The inset exhibits the extracted squeezing rates as a function of the displacement amplitude $|\beta|^2$, which agree well with the simulation results (dashed line). The solid line represents the ideal squeezing rate of $K\beta^2$. **d** Measured 2D Wigner functions of the generated squeezed states for various Trotter steps with $|\beta|^2 = 100$. The extracted state fidelities and squeezing levels are indicated accordingly.

the number of Trotter steps with a fixed $N_{\mathrm{meas}} = 3.2 \times 10^4$. The results indicate that as the number of Trotter steps increases, the Allan deviation for displacement measurement significantly decreases, highlighting the large metrological advantage of these nonclassical states in surpassing the standard quantum limit.

## Discussion

In conclusion, we have experimentally demonstrated displacement-enhanced quantum squeezing amplification with a weak Kerr non-linear oscillator in a superconducting microwave cavity, leveraging the Trotterization technique. This method distinguishes itself from previous displacement-enhanced operations[26,43] and other squeezing approaches[24–28] by eliminating decoherence errors induced by the ancillary qubit and suppressing cavity dephasing errors through echoed displacements during the evolution. Our deterministic squeezing approach ultimately generates nonclassical microwave photonic states with a maximum squeezing degree of 14.6 dB over quasiclassical coherent states. The squeezing degree is primarily limited by the displacement drive amplitude due to electronic hardware constraints and the breakdown of the dispersive approximation (see Supplementary Note 6). These large squeezed states are demonstrated to exhibit large metrological advantages, promising potential applications in quantum-enhanced metrology[50]. Furthermore, our approach provides a hardware-efficient way to engineer strong two-photon squeezing operations, which is beneficial for quantum information processing with Kerr cat qubits[46–48], bosonic quantum chemistry simulations[57], and topological phase transitions[51,52], as well as for

reducing decoherence of quantum systems[14,58,59]. A recent notable advancement in quantum squeezing with acoustic wave resonators[32] suggests the potential for easily adapting our approach to achieve even larger squeezing with microwave phonons, photons, or magnons in qubit-oscillator systems[29,60–62].

## Methods

### Characterization of squeezing parameters

The generated squeezed states are characterized by comparing measured Wigner functions to those of ideal squeezed states $|\xi\rangle$. The theoretical 2D Wigner function for an ideal squeezed state is given by $W(\alpha) = \frac{2}{\pi} \exp(-2|\nu|^2)$, where $\nu = \cosh(|\xi|)\alpha^* + \sinh(|\xi|)\alpha$ and $\alpha$ is the complex phase-space variable. The measured 2D Wigner functions in Figs. 3d and 2b, c are fitted directly with this expression to extract the squeezing parameter. However, measuring the full 2D Wigner function is time-consuming. Therefore, we employ an efficient approach by measuring only 1D Wigner cuts along the squeezing quadrature ($\mathrm{Im}(\alpha) = 0$) and the anti-squeezing quadrature ($\mathrm{Re}(\alpha) = 0$), then performing a global fit to these two traces to efficiently extract the squeezing parameter $|\xi|$ (see Fig. 3b). See the Supplementary Note 3 for detailed descriptions.

### Extraction of Fisher information

In quantum metrology experiments, we employ the FI to quantify the precision achievable in estimating the displacement parameter $\alpha$ using nonclassical squeezed states. The classical FI is defined as $I_{\mathrm{F}}(\alpha) = \sum_\mu \frac{1}{P_\mu} \left( \frac{\partial P_\mu}{\partial \alpha} \right)^2$[56], where $P_\mu$ denotes the probability distribution

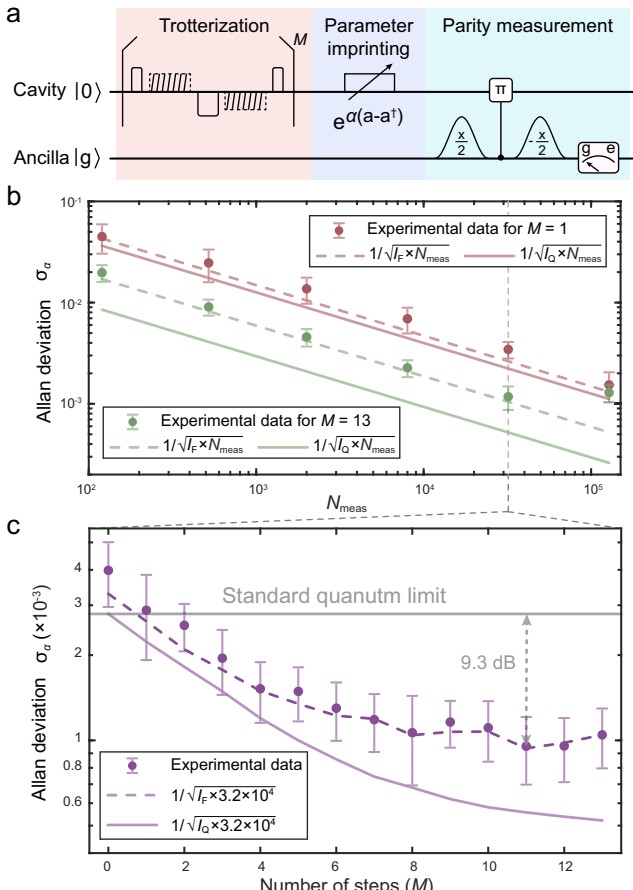

**Fig. 4 | Quantum metrology for measuring small displacements using the generated squeezed states. a** Experimental sequence for the displacement measurement. **b** Allan deviation for estimating the displacement amplitude using squeezed states generated with different Trotter steps (red: $M = 1$, green: $M = 13$) versus the number of independent measurements $N_{meas}$. The increase of the last point for $M = 13$ may primarily be attributed to displacement drifts in the experimental apparatus. **c** Allan deviation for estimating the displacement amplitude as a function of Trotter steps for generating the squeezed states with a fixed $N_{meas} = 3.2 \times 10^4$. Solid (dashed) line represents the theoretical Cramér-Rao bound using quantum (classical) FI of the squeezed states. Error bars are standard deviations from repeated experiments.

from a set of measurement projections $M_\mu$. In our experiment, the measurement performed on the ancillary qubit yields only two outcomes, $\mu = \{g, e\}$, corresponding to the qubit in ground and excited states. Consequently, the classical FI simplifies to $I_F(\lambda) = \frac{1}{P_g(1-P_g)} \left( \frac{dP_g}{d\lambda} \right)^2$. For an ideal squeezed state $|\xi\rangle$, the quantum FI is calculated by $I_Q = 4(\langle \xi | h_\alpha^2 | \xi \rangle - \langle \xi | h_\alpha | \xi \rangle^2) = 4e^{2|\xi|}$, with $h_\alpha = i(a - a^\dagger)$ representing the interrogation Hamiltonian that encodes the displacement parameter via the unitary transformation $U_\alpha = \exp(-i\alpha h_\alpha)$ (assuming $\alpha$ is real). These FI quantities are directly related to the variance of the estimated displacement amplitude through the Cramér-Rao inequality[56]: $\Delta \alpha \geq \frac{1}{\sqrt{N_{meas} I_F}} \geq \frac{1}{\sqrt{N_{meas} I_Q}}$.

**Bayesian inference**

To estimate the displacement parameter $\alpha$ from the measured qubit population, we employ the Bayesian inference approach by maximizing the posterior probability $P(\alpha | M) \propto P(\alpha) \times P(M | \alpha)$, where $P(M | \alpha)$ is the likelihood of observing the ancilla in the ground state ($M = g$) or excited state ($M = e$) under $\alpha$. Assuming a uniform prior probability $P(\alpha)$ over $\alpha \in [0, 0.5]$, Bayesian estimation reduces to maximum likelihood

estimation. The displacement amplitude is estimated by maximizing the log-likelihood function $L(M | \alpha) = \ln[P(M | \alpha)] = P_g^{exp} \ln[P_g(\alpha)] + (1 - P_g^{exp}) \ln[1 - P_g(\alpha)]$ via $\alpha_{est} = \arg\max_\alpha L(M | \alpha)$, incorporating the measured qubit ground state population $P_g^{exp}$ and fitted qubit ground state probability distribution $P_g(\alpha)$ (see Supplementary Note 5).

## Data availability

Source data are provided with this paper, and more detailed data are available from the corresponding authors upon request. Source data are provided in this paper.

## Code availability

The code used for this study is available from the corresponding authors upon request.

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

## Acknowledgements

We thank Zhen-Biao Yang and Xiu Gu for valuable suggestions related to this work. This work was supported by the Quantum Science and Technology-National Science and Technology Major Project (Grants No. 2024ZD0302300, No. 2021ZD0301703), the Guangdong Basic and Applied Basic Research Foundation (Grant No. 2024B1515020013), the National Natural Science Foundation of China (Grants No. 12422416, No. 12274198, No. 12575025), the Shenzhen Science and Technology Program (Grant No. RCYX20210706092103021), the Guangdong Provincial Key Laboratory (Grant No. 2019B121203002), the Shenzhen-Hong Kong cooperation zone for technology and innovation (Contract No. HZQB-KCZYB-2020050).

## Author contributions

Y.X. conceived the idea and designed the experiment. Y.C. and X.D. performed the experiments, data analysis, and numerical simulations under the supervision of Y.X. L.Z. fabricated the superconducting qubit chip assisted by S.L. Z.N., J.M., P.Z., and L.H. contributed to the experimental setup, and Y.C., X.D., P.H., and Y.X. discussed the results. Y.C. and Y.X. wrote the manuscript with input from all authors. Y.X. and D.Y. supervised the project.

## Competing interests

The authors declare no competing interests.
