## [Transparent Peer Review file · Nature Communications]

Quantum squeezing amplification with a weak Kerr nonlinear oscillator

Corresponding Author: Dr Yuan Xu

Version 0:

Reviewer comments:

Reviewer #1

(Remarks to the Author)
see file attached

Reviewer #2

(Remarks to the Author)

I enjoyed this manuscript's clever use of a weak Kerr nonlinearity plus an off-resonant drive to generate and amplify squeezing inside a superconducting microwave cavity, reaching 14.6 dB of intracavity squeezing. The idea of cycling a coherent state in a displaced frame to avoid collapse, then using a Trotterized echo of alternating displacement phases to reinforce two-photon squeezing while canceling unwanted Kerr rotations, is both elegant and original. It's worth adding a brief, intuitive note on "photon blockade" (the Kerr-induced photon-photon interaction) and how the drive detuning neutralizes it at cycle endpoints, and a one-sentence plain-English description of the Trotterization, alternating displacement phases so that nonlinear phase errors cancel while squeezing terms accumulate. It would also be more informative to give T_2^* or T_2 echo times in the main text instead of only the pure dephasing time. Speaking of limits, the text notes that "the squeezing degree can be further enhanced by engineering a weaker Kerr nonlinearity and a larger displacement amplitude," but it's unclear what the practical ceiling is; a short discussion of realistic bounds—perhaps set by calibration fidelity, drive amplitude constraints, or decoherence rates—would be a nice addition. A sentence contrasting this intracavity approach with standard JPAs or other methods would help to contrast with other squeezing work. Despite these minor points, the experiments are convincing and I recommend publication in Nature Communications.

Reviewer #3

(Remarks to the Author)

Referee report for "Quantum squeezing amplification with a weak Kerr nonlinear oscillator"

This work outlines two different approaches for generating squeezed states, both of which utilize the inherent non-linearity associated with a bosonic mode coupled to a non-linear element, a transmon qubit in this case. Squeezed states have many applications, especially in sensing due to their asymmetry of sensitivity along the two different quadratures. The generation of highly squeezed states can be difficult, and this manuscript proposed and implemented a novel protocol to accomplish record degree of squeezing that leverages a very commonplace device technology/architecture to the field of circuit QED, making the work broadly applicable to the field. Thus, I believe the experiments presented are clever and useful along with being of interest to a relatively large reader base in the field. However, I have some issues with the presentation of the work as written.

The first major issue is the lack of intuitive/heuristic explanation of the experiment/results. The layout of the manuscript is to propose the model in the form of a Hamiltonian, show the results and then show agreement with a numerical simulation of the model. This does not really provide a nice intuitive/heuristic picture of the underlying physics for the reader to understand. This issue is most prominent in the discussion of the driven case. The oblong trajectory shown in the cartoon in fig. 1c) and the data in fig. 2 b) is rather unique/interesting. It is not immediately clear what the heuristic reason for the shape

of this trajectory is and the importance of the drive parameters. I believe providing some more intuitive explanation for this would be very useful. It seems that there should be a clear intuitive picture looking at the drive as pulling the state along the negative $\text{Im}(\alpha)$ axis while the Kerr term is rotating the state around with a frequency proportional to the average photon number. This should also explain how the drive prevents a full rotation in phase space, leading to the squeezing effect from the Kerr before it fully evolves to the point where the squeezing effect is ruined.

The second major issue is the unclear relationship between the two techniques presented, the driven approach and the Trotterization technique. Currently the manuscript introduces the driven approach as the main focus, while the Trotterization technique appears to be an extension. However, the Trotterization approach appears superior as it achieves stronger and faster squeezing. Furthermore, it does not have to deal with a brute-force-optimized external drive to perform and is generally easier to understand. The manuscript would be better served by highlighting the Trotterization technique and treating the driven method as an extension instead. If there are some advantages/disadvantages to the two approaches that make them more or less useful in different applications, this should be discussed to give clear motivation for including the different approaches. This explanation/comparison can then be incorporated into a more cohesive transition between the sections of the two methods which would tie the paper together much more nicely and give clearer motivation for what is presented.

Furthermore, there are some areas where further discussions or revisions are needed:

In the discussion on page 4 of the Trotterization technique, the manuscript mentions that the detuned drive is unnecessary so $\Omega_d=0$ but then it is mentioned that this allows one to “engineer the frequency detuning to mitigate Kerr-induced phases”. This is confusing as I don’t understand what a frequency detuning from a drive means if the drive has 0 amplitude. Again, a more intuitive explanation of the drive would be helpful to understand the differences in this technique versus the first one and would allow for more of a smooth segue in the paper between topics. It seems that the drive in the first technique is what prevents the full Kerr rotation, only allowing a part of the evolution to occur where the squeezing happens whereas in the Trotterization method, the rapid switching between opposite signed displaced states prevents any single step of Kerr evolution from going beyond just the squeezing stages but this should be thought out and more explicitly explained in the paper.

Since all the squeezed states generated are warped to a degree, or deviated from an ideal squeezed state, it would be nice to add a note about how this would affect their use in detection applications regarding potential detriments this effect has.

The fit of $K^{(-1/3)} \Omega^{(-2/3)}$ is in very good agreement with the data in the supplementary, it seems that there should be a clear derivation for this. I think some more work to derive or understand this relation would be useful. At the very least, simulations with a broader parameter spectrum should be performed so the limits of applicability of this expression can be confidently explained.

In section IV of the supplementary, the detuning of the drive is again discussed as a tuned parameter but here it is in relation to the Trotterization technique where there is no drive and I don’t understand what the meaning of detuning has with respect to a drive with no amplitude that does not play a role in the process. Please explain this in the text.

In the abstract, the first line states, “Quantum squeezed states, with reduced quantum noise...”. The word “reduced” here does not seem accurate and “biased noise” seems to be a more appropriate term as the noise will be reduced in one quadrature but enhanced in another for a squeezed state.

Excessively self-congratulatory adjectives like “ingenious” (near the end of the introduction section) is not appropriate for a scientific publication.

In the concluding paragraph, the authors made a strong claim without evidence that “the squeezing degree can be further enhanced by engineering a weaker Kerr nonlinearity and a larger displacement amplitude”. Weaker single-photon nonlinearity but a larger displacement does not necessarily mean weaker spurious higher-order terms. Furthermore, as the displacement amplitude increases and the average photon number becomes larger, would the increased photon loss rate cause problems for this scheme? The authors must provide much more in-depth analysis or even experimental tests to back this claim up. Otherwise, this claim should be removed.

Final recommendation:

Overall, the experiments performed and the approach to squeezed state generation are very nice and innovative and would find useful applications in the field. However, the presentation of the results, as written, needs some work. At the moment I cannot recommend publication but if the above-mentioned problems are fixed, I would recommend publication.

Reviewer #4

(Remarks to the Author)

Reviewer #5

(Remarks to the Author)

Version 1:

Reviewer comments:

Reviewer #1

(Remarks to the Author)

Reviewer #2

(Remarks to the Author)

I think the paper can now be published

Reviewer #3

(Remarks to the Author)

The additions made to the text were very good overall and the manuscript has really been elevated from the first draft. The addition of an intuitive picture to the results was very helpful in understanding the experiment. The addition of the metrology experiment gave much better motivation for the experimental technique and put the claims on much more solid footing. The derivation of the power law in the supplementary was very useful. A lot of wording and confusing explanations were cleared up, making the paper much more cohesive and understandable.

There are still three points we think should be addressed:

1. As we noted earlier, the Trotterization approach seemed easier to implement, achieved better results faster and was more intuitive to understand, while the driven approach seemed like more of a side-show and has no real advantage. We were therefore confused about the strong emphasis on the driven approach in the writing. This does not impact the overall significance/correctness of the work, but it just seems an awkward way to present the results. The authors added some sentences in the revision to segue these results a bit nicer, but didn't really justify this approach of presentation. If there is some distinct advantage to the driven approach, it should be made more clear.

2. There still seems to be insufficient evidence to support the claim that the squeezing degree can be increased with a larger displacement amplitude and weaker Kerr. The authors added an explanation for this statement with a simulation showing that with a larger displacement and smaller Kerr, the simulated fidelity is greater than 99%. However, it is stated that a predominant source of errors is the higher order Kerr terms such as 6th and 8th order terms. It would seem that these higher order terms scale such that they would have an even larger contribution as photon numbers get larger, making them a dominant error pathway in this discussion, so using a simulation with just the 4th order Kerr terms and single photon loss seems to be neglecting a very large error avenue for the statement that is trying to be made here. Either the higher order terms should be taken into account and shown concretely to still allow for higher fidelities, or this statement should be taken out of the text.

3. The additional experiments and writeup on the newly added Quantum metrology section are a great addition to the paper and make the message much more convincing. The only point to add here is an explanation for the data point in Fig. 5b) for the $m=13$ point at the highest N_{meas} would be nice, it is odd that this point is so far off the trend line and outside of any statistical fluctuation based on the size of the error bars.

With resolution of these points, we would recommend publication.

Reviewer #4

(Remarks to the Author)

Reviewer #5

(Remarks to the Author)

Version 2:

Reviewer comments:

Reviewer #1

(Remarks to the Author)

We thank the authors for their response to our remaining technical questions. All our questions are fully resolved now and we recommend publication.

Reviewer #3

(Remarks to the Author)

My co-reviewer and I have read the revised manuscript, and we agree that the remaining issues of the manuscript have been address adequately. We recommend publication of the manuscript in Nature Communications.

Reviewer #4

(Remarks to the Author)

Reviewer #5

(Remarks to the Author)

Response to Reviewers #1 & #4's Comments:

Comment 1:

In this article, the authors presented a scheme to squeeze vacuum and Fock states in a superconducting microwave cavity. They use the weak Kerr non-linearity of the cavity with an off-resonant drive. The resulting action resembles a squeezing operation, whose effective strength is enhanced during the evolution by the strong displacement in phase space. To mitigate the effect of the unwanted term in the Hamiltonian (second line in Eq.2), the authors employed Trotterization method. The experimental results in the paper show states that resemble highly squeezed vacuum and Fock states.

The positive aspects of this papers are:

- 1. The method does not use the qubit in the evolution, which makes it not prone to qubit decoherence.*
- 2. The experimental technique is simple and only uses natively available gates and dynamics. Thus, it is easily implemented on typical bosonic cQED setups.*

Response:

We thank the Reviewer for the nice summary and positive assessments of our method, including its robustness against qubit decoherence and experimental simplicity. These novel features have also been recognized and highlighted by other reviewers. Following the Reviewer's suggestions, we have included additional experiments and clarifications throughout the revised manuscript. We believe these major revisions would significantly strengthen our manuscript and are confident that it now meets the high standards for publication in *Nature Communications*.

Comment 2:

The authors only presented the degree of squeezing as a figure of merit. To us, this does not capture the full picture. The states visually look at squeezed states but have many artefacts, as the Kerr dynamics is not the same as squeezing. So to claim that strongly squeezed states are produced seems to oversimplify what is going on. What about the overall quality of the generated states, i.e., the fidelity of the generated state to the target squeezed states?

Response:

We thank the Reviewer for raising this point. In our manuscript, we have shown the squeezing degree as a figure of merit to quantify the quantum sensing capability of the generated squeezed states. Actually, we have presented the fidelities of these states to ideal squeezed

states in Fig. S13 in the Supplementary Material. These fidelities are extracted from the reconstructed density matrices via Wigner tomography, achieving a maximum squeezing degree of 14.6 dB with a fidelity exceeding 0.5. To provide a clearer picture of the generated states' quality, we have now included the state fidelity alongside the squeezing degree in Fig. 4(d) in the main text of the revised manuscript.

Comment 3:

The description of the analysis is also quite vague. They mention that they perform fit to extract the squeezing level. What about the fitting errors? This is particularly a good thing to show as one can achieve high squeezing level but the state smears out in phase space (like in Fig. 4d $M=13$). What about the anti-squeezing level on these states? Do they make sense? It is not complete to put maximum squeezing 14.6 dB in the abstract without also presenting how good the quality of the state is.

Response:

We appreciate the Review's insightful feedback. In our manuscript, the fitting errors for the squeezing level, obtained from both the 1D and 2D Wigner function fits, are evaluated using 95% confidence intervals and are represented by the error bars in Fig. 3(d) and Fig. 4(c) in the main text. These fits are performed globally to account for both the squeezing and anti-squeezing quadratures equally.

To separately quantify the anti-squeezing level, we also fit individual orthogonal quadrature cuts of the measured Wigner function. The results, shown in the following Figure 1 [see Fig. S8(d) in the revised Supplementary Material], are consistent with those obtained from the global 2D Wigner fits, validating the effectiveness and reliability of the fitting methodology.

Figure 1: Extracted squeezing and anti-squeezing parameters from individual fits as a function of the number of Trotter steps. Dashed lines correspond to the squeezing parameters obtained from global 2D Wigner fits.

The quality of the generated states is quantified by calculating the state fidelities between the reconstructed density matrices and ideal squeezed states. These values are now included in Fig. 4(d) in the main text and Fig. S13 in the Supplementary Material of the revised manuscript.

Comment 4:

The authors mentioned that at the end they displaces back the compressed coherent state back to origin and perform a virtual phase rotation to eliminate rotation angle. What is then the role of the cycle, as one can stop at any point in Fig. 2b and performed the mentioned corrections?

Response:

We agree with the Reviewer that the phase correction operation could, in principle, be applied at any point during the evolution to generate squeezed states. This approach was indeed used to track the state trajectory dynamics during the cyclic evolution, as shown in Fig. 2b in the main text. However, to achieve optimal state fidelity, we specifically select the evolved state at the end of each evolution cycle. To clarify this point, we have performed numerical simulations of the cyclic evolution dynamics and examined both the state fidelity and squeezing parameter as a function of the evolution time, as shown in the following Figure 2 [see Fig. S2(d) in the revised Supplementary Material]. The results indicate that the fidelity reaches a maximum at the end of each cycle, and the squeezing level is enhanced by increasing the number of cycles. Therefore, we choose to perform phase correction rotation on the evolved state at the end of each cycle to maximize the fidelity of the generated squeezed states. In response, we have added an initiative explanation in the fourth paragraph on page 2 in the main text and Fig. S2(d) in the Supplementary Material of the revised manuscript.

Figure 2: Numerically simulated state fidelity and squeezing parameter as a function of the evolution time. Dashed vertical lines correspond to the final moment of each evolution cycle.

Comment 5:

Ideally, squeezed vacuum does not have Wigner negativity, it is simply a vacuum state with positive Wigner function squeezed in one quadrature. The Wigner logarithmic negativity that

the authors computed are caused by imperfections and does not show the main non-classical feature of the squeezed state.

Response:

We thank the Reviewer for raising this point. We agree that an ideal squeezed vacuum state is a Gaussian state and exhibits no Wigner negativity. In our experiment, the observed Wigner negativity of the generated states arises from the evolution of the Kerr parameter oscillator Hamiltonian, as also evidenced in Ref. [Nat. Phys. 20, 1448 (2024)]. The coexistence of squeezing and non-Gaussian features may be beneficial for certain quantum information applications. To avoid any misunderstanding, we have therefore removed the discussions of Wigner negativity throughout the revised manuscript and focused instead on the squeezing amplification and quantum metrology application using the generated nonclassical states.

Comment 6:

From what is presented in the manuscript, there is little evidence that the states created are close to ideal squeezed states. Given this, are the reported states actually useful for anything that a typical squeezed vacuum is suited for? The last figure computes the FI, but it is not in the context of any real tasks that uses the state (such as metrology). It does not seem compelling or obvious to us that such states are useful given their pronounced artefacts and deviations from ideal squeezing. To show us, the authors would need to perform a real experiment using this state, which I think would require a rather major revision.

Response:

We appreciate the Reviewer's constructive suggestion and make a major revision to perform a quantum sensing experiment using the generated nonclassical states.

To evaluate how closely our generated states resemble ideal squeezed states, we have calculated the state fidelities using reconstructed density matrices, as presented in Fig. S13 of the Supplementary Material.

Following the Reviewer's suggestion, we have performed additional experiments to demonstrate the practical utility of these generated states in quantum metrology. Specifically, we have conducted a displacement sensing task in which the generated squeezed states are used to estimate small displacements along the position quadrature. The experimental details of the procedure, analysis, and results are presented in the following Figure 3 [see the newly added Fig. 5 in the main text] and Section V in the Supplementary Material of the revised manuscript.

In this experiment, we first generate the squeezed states with varying Trotter steps (M), then imprint a small displacement parameter, and finally perform a parity measurement. Using Bayesian inference, we extract the Allan deviations for displacement estimation as a function of both the number of independent measurements (N_{meas}) and Trotter steps (M). The results indicate that the displacement estimation precision improves as the number of Trotter steps or the number of experiments increases. We achieve a maximum metrological gain of 9 dB beyond the standard quantum limit at $M = 12$. These experimental results confirm that our generated squeezed states remain highly useful for practical metrology applications, despite phase-space twisting and deviations from ideal squeezing. We believe these major revisions would significantly strengthen our manuscript and meet the high standards for publication in *Nature Communications*.

Figure 3: Quantum metrology for measuring displacements using generated squeezed states. (a) Experimental sequence. (b) Extracted Allan deviation for displacement measurements as a function of the number of experiments. (c) Extracted Allan deviation as a function of the number of Trotter steps for generating the squeezed states.

Response to Reviewer #2's Comments:

Comment 1:

I enjoyed this manuscript's clever use of a weak Kerr nonlinearity plus an off-resonant drive to generate and amplify squeezing inside a superconducting microwave cavity, reaching 14.6 dB of intracavity squeezing. The idea of cycling a coherent state in a displaced frame to avoid collapse, then using a Trotterized echo of alternating displacement phases to reinforce two-photon squeezing while canceling unwanted Kerr rotations, is both elegant and original.

Response:

We sincerely appreciate Reviewer #2's thorough summary of our key results, including cyclic squeezing dynamics and Trotterized echo squeezing amplification. We are particularly gratified by the recognition of our approach as "clever" and "elegant and original".

Comment 2:

It's worth adding a brief, intuitive note on "photon blockade" (the Kerr-induced photon–photon interaction) and how the drive detuning neutralizes it at cycle endpoints, and a one-sentence plain-English description of the Trotterization, alternating displacement phases so that nonlinear phase errors cancel while squeezing terms accumulate.

Response:

We thank the Reviewer for this valuable suggestion. In the displacement frame of the driven Hamiltonian, the photon blockade term comes from the combination of the Kerr nonlinearity and the detuned drive, as also evidenced in Ref. [Sci. Adv. 7, eabj1916 (2021)]. By choosing appropriate drive strength and detuning, these two effects result in cyclic state trajectory dynamics, achieving a maximum squeezing fidelity at the endpoint of each cycle. Following the Reviewer's suggestion, we have added an intuitive note: "Intuitively, the photon blockade term in the displaced frame arises from the interplay between the detuned drive and the Kerr nonlinearity. The detuned drive acts as a driving force pulling the coherent state along the negative imaginary axis in phase space, while the Kerr term rotates the state anticlockwise at a rate proportional to the average photon number. Carefully balancing these two effects prevents state collapse and enables periodic cycling trajectory dynamics (see Supplement). The mirror-symmetry trajectory around the real axis in phase space effectively cancels the photon blockade effect at the end of each cycle, achieving the displacement-enhanced squeezing operation with a squeezing rate $K\beta^2$, as indicated by the two-photon squeezing term in Eq. (2)." in the fourth paragraph on page 2 in the main text of the revised manuscript. We have also provided a more detailed discussion and derivation of the cyclic trajectory

dynamics in Sections II-B and II-C of the revised Supplementary Material.

According to the Reviewer's suggestion regarding the Trotterization method, we have also added a one-sentence description: "*This Trotterization method alternates the phases of the displacement frame, which effectively cancels the undesired nonlinear photon-blockade phase errors while reinforcing the two-photon squeezing term.*" in the first paragraph on page 4 in the main text of the revised manuscript.

Comment 3:

It would also be more informative to give T_2^ or T_2 echo times in the main text instead of only the pure dephasing time.*

Response:

Thanks for this suggestion. We have added the Ramsey and echo decoherence times in the main text of the revised manuscript.

Comment 4:

Speaking of limits, the text notes that "the squeezing degree can be further enhanced by engineering a weaker Kerr nonlinearity and a larger displacement amplitude," but it's unclear what the practical ceiling is; a short discussion of realistic bounds—perhaps set by calibration fidelity, drive amplitude constraints, or decoherence rates—would be a nice addition

Response:

We appreciate Reviewer #2's valuable suggestion. The demonstrated Trotterized squeezing approach indicates that the squeezing degree is enhanced by increasing the displacement amplitudes. However, in our realistic experiment, the achievable displacement amplitudes are primarily limited due to electronic hardware constraints and the breakdown of the dispersive approximation at high photon numbers, which would induce parasitic qubit excitations during the evolution. The intrinsic Kerr nonlinearity of the cavity would also restrict the achievable photon number and introduce phase-space twisting. To generate a larger squeezing degree with high fidelity, one could engineer a weaker Kerr nonlinearity, allowing larger displacements without disrupting the dispersive approximation.

In order to clarify this point, we have performed numerical simulations, and the results are presented in Section VII and Fig. S13 in the revised Supplementary Material. The simulation results indicate that employing a weaker Kerr nonlinearity $K/2\pi = 1$ kHz would yield a state fidelity exceeding 0.99 with a squeezing parameter $|\xi| \approx 1.5$. In response to the Reviewer's suggestion, we have revised the relevant sentence in the last paragraph to: "*The squeezing*

degree is primarily limited by the displacement drive amplitude due to electronic hardware constraints and the breakdown of the dispersive approximation.” to discuss the realistic bounds of our squeezing approach. A more detailed error analysis is also provided in Section VII of the revised Supplementary Material.

Comment 5:

A sentence contrasting this intracavity approach with standard JPAs or other methods would help to contrast with other squeezing work.

Response:

We thank Reviewer #2 for this suggestion. In the revised manuscript, we have added a brief description, “This method distinguishes itself from previous displacement-enhanced operations and other squeezing approaches by eliminating decoherence errors induced by the ancillary qubit and suppressing cavity dephasing errors through echoed displacements during the evolution.” in the concluding paragraph in the main text to contrast our approach to other squeezing works. A more detailed comparison and analysis is provided in Section VIII of the revised Supplementary Material.

Comment 6:

Despite these minor points, the experiments are convincing and I recommend publication in Nature Communications.

Response:

We appreciate Reviewer #2’s recognition of our convincing experiments and are highly encouraged by the recommendation for publication in *Nature Communications*. We have clearly addressed all the minor points raised by the Reviewer in the revised manuscript.

Response to Reviewers #3 & #5's Comments:

Comment 1:

This work outlines two different approaches for generating squeezed states, both of which utilize the inherent non-linearity associated with a bosonic mode coupled to a non-linear element, a transmon qubit in this case. Squeezed states have many applications, especially in sensing due to their asymmetry of sensitivity along the two different quadratures. The generation of highly squeezed states can be difficult, and this manuscript proposed and implemented a novel protocol to accomplish record degree of squeezing that leverages a very commonplace device technology/architecture to the field of circuit QED, making the work broadly applicable to the field. Thus, I believe the experiments presented are clever and useful along with being of interest to a relatively large reader base in the field. However, I have some issues with the presentation of the work as written.

Response:

We sincerely thank the Reviewer for their thorough summary and positive assessment of our work. We are particularly encouraged by their recognition of the novelty and practical relevance of our approach, as well as its potential interest to the circuit QED community. We also appreciate the constructive suggestions aimed at improving the clarity and presentation of the manuscript. In the following, we address each comment individually.

Comment 2:

The first major issue is the lack of intuitive/heuristic explanation of the experiment/results. The layout of the manuscript is to propose the model in the form of a Hamiltonian, show the results and then show agreement with a numerical simulation of the model. This does not really provide a nice intuitive/heuristic picture of the underlying physics for the reader to understand. This issue is most prominent in the discussion of the driven case. The oblong trajectory shown in the cartoon in fig. 1c) and the data in fig. 2 b) is rather unique/interesting. It is not immediately clear what the heuristic reason for the shape of this trajectory is and the importance of the drive parameters. I believe providing some more intuitive explanation for this would be very useful. It seems that there should be a clear intuitive picture looking at the drive as pulling the state along the negative $\text{im}(\alpha)$ axis while the Kerr term is rotating the state around with a frequency proportional to the average photon number. This should also explain how the drive prevents a full rotation in phase space, leading to the squeezing effect from the Kerr before it fully evolves to the point where the squeezing effect is ruined.

Response:

We appreciate the Reviewer for pointing out this issue. We agree that the cyclic squeezing trajectory of our experimental results is rather unique and interesting, and providing a clearer intuitive picture significantly strengthens the manuscript. As suggested by the Reviewer, we have added an intuitive explanation: “Intuitively, the photon blockade term in the displaced frame arises from the interplay between the detuned drive and the Kerr nonlinearity. The detuned drive acts as a driving force pulling the coherent state along the negative imaginary axis in phase space, while the Kerr term rotates the state anticlockwise at a rate proportional to the average photon number. Carefully balancing these two effects prevents state collapse and enables periodic cycling trajectory dynamics (see Supplement). The mirror-symmetry trajectory around the real axis in phase space effectively cancels the photon blockade effect at the end of each cycle, achieving the displacement-enhanced squeezing operation with a squeezing rate $K\beta^2$, as indicated by the two-photon squeezing term in Eq. (2).” in the fourth paragraph on page 2 in the main text of the revised manuscript. We have also provided a more detailed discussion and derivation of the cyclic trajectory dynamics in Sections II-B and II-C of the revised Supplementary Material.

Comment 3:

The second major issue is the unclear relationship between the two techniques presented, the driven approach and the Trotterization technique. Currently the manuscript introduces the driven approach as the main focus, while the Trotterization technique appears to be an extension. However, the Trotterization approach appears superior as it achieves stronger and faster squeezing. Furthermore, it does not have to deal with a brute-force-optimized external drive to perform and is generally easier to understand. The manuscript would be better served by highlighting the Trotterization technique and treating the driven method as an extension instead. If there are some advantages/disadvantages to the two approaches that make them more or less useful in different applications, this should be discussed to give clear motivation for including the different approaches. This explanation/comparison can then be incorporated into a more cohesive transition between the sections of the two methods which would tie the paper together much more nicely and give clearer motivation for what is presented.

Response:

We thank the Reviewer for this insightful comment. We agree with the Reviewer that the Trotterization approach offers significant advantages, including stronger and faster squeezing, and simpler and easier implementation without the need for brute-force optimization. In our manuscript, the driven approach serves to illustrate the fundamental cyclic squeezing

dynamics under the detuned driven Kerr oscillator Hamiltonian, which forms the basis for introducing the digital Trotterization squeezing approach. To better motivate and contrast these two methods, we have added sentences: "*The quantum dynamics of the driven Kerr Hamiltonian with engineered parameters can be employed to generate two-photon squeezing, which can be further amplified through the Trotterization technique.*" in the third paragraph on page 2 and "*To further enhance the squeezing levels without relying on brute-force optimizations, we propose utilizing the Trotterization technique to eliminate the residual photon-blockade effect in Hamiltonian Eq. (2). This Trotterization strategy is based on the Kerr squeezing dynamics mentioned above, enabling a digital squeezing protocol that relies solely on simple, natively available gates and dynamics.*" in the third paragraph on page 3 in the revised main text. These revisions provide a clearer comparison of the two strategies and emphasizes the practical benefits of the Trotterization technique, thus improving the transition between the two sections to enhance the overall cohesion of the manuscript.

Comment 4:

Furthermore, there are some areas where further discussions or revisions are needed: In the discussion on page 4 of the Trotterization technique, the manuscript mentions that the detuned drive is unnecessary so $\Omega_d=0$ but then it is mentioned that this allows one to "engineer the frequency detuning to mitigate Kerr-induced phases". This is confusing as I don't understand what a frequency detuning from a drive means if the drive has 0 amplitude. Again, a more intuitive explanation of the drive would be helpful to understand the differences in this technique versus the first one and would allow for more of a smooth segue in the paper between topics. It seems that the drive in the first technique is what prevents the full Kerr rotation, only allowing a part of the evolution to occur where the squeezing happens whereas in the Trotterization method, the rapid switching between opposite signed displaced states prevents any single step of Kerr evolution from going beyond just the squeezing stages but this should be thought out and more explicitly explained in the paper.

Response:

We thank the Reviewer for pointing out this issue and apologize for the confusion. We agree with the Reviewer that for the driven squeezing approach, the detuned drive serves as a driving force to prevent the full Kerr rotation and only allows a part of evolution to occur, while for the Trotterization technique, the rapid switching between opposite-signed displaced states prevents any single step of Kerr evolution from going beyond the squeezing stages, thus rendering the detuned drive unnecessary during the evolution. Therefore, in our Trotter

experiment, we set $\Omega_d = 0$ and apply a virtual phase rotation (effectively emulating a frequency detuning during the evolution) on the cavity after each Kerr evolution to compensate for Kerr-induced phase accumulation. To avoid any confusion and clarify this point, we have revised the relevant sentence in the first paragraph on page 4 to “*Note that in this scenario, the rapid switching between opposite-signed displaced states prevents any single step of Kerr evolution from going far beyond the squeezing stages, rendering the detuned drive unnecessary during the Kerr evolution. We therefore set $\Omega_d = 0$ to restrict the average photon numbers of the intermediate states and engineer a virtual phase shift after each evolution cycle to mitigate the Kerr-induced phase on the squeezed state (see Supplement).*” in the revised manuscript.

Comment 5:

Since all the squeezed states generated are warped to a degree, or deviated from an ideal squeezed state, it would be nice to add a note about how this would affect their use in detection applications regarding potential detriments this effect has.

Response:

We appreciate the Reviewer for this valuable suggestion. Although the generated squeezed states are warped and deviated from an ideal squeezed state, they still provide promising metrological advantages in quantum sensing applications.

To clarify this point and directly assess the practical utility of these warped squeezed states, we have conducted additional experiments to sense small displacements along the position quadrature. The experimental details of the procedure, analysis, and results are presented in Fig. 5 in the main text and Section V in the Supplementary Material of the revised manuscript. In this experiment, we first generate the squeezed states with varying Trotter steps (M), then imprint a small displacement parameter, and finally perform a parity measurement. Using Bayesian inference, we extract the Allan deviations for displacement estimation as a function of both the number of independent measurements (N_{meas}) and the number of Trotter steps (M). The results indicate that the displacement estimation precision improves as the number of Trotter steps or the number of experiments increases. We achieve a maximum metrological gain of 9 dB beyond the standard quantum limit at $M = 12$. The difference between the measurement results and that from ideal squeezed states arises from the infidelity of the generated squeezed state. Despite the deviations from ideal squeezing, these experimental results confirm that our generated squeezed states remain highly useful for practical metrology applications.

Comment 6:

The fit of $K^{(-1/3)} \Omega^{(-2/3)}$ is in very good agreement with the data in the supplementary, it seems that there should be a clear derivation for this. I think some more work to derive or understand this relation would be useful. At the very least, simulations with a broader parameter spectrum should be performed so the limits of applicability of this expression can be confidently explained.

Response:

We thank the Reviewer for this suggestion. In response, we have added a new subsection II-C in the revised Supplementary Material that provides a detailed analytical derivation of the relation $K^{(-1/3)} \Omega^{(-2/3)}$ using a semiclassical approximation of the Heisenberg equations of motion during the evolution. Additionally, we perform systematic numerical simulations across a broader parameter range to further verify the applicability of the expression. The results, presented in Fig. S4 in the revised Supplementary Material, confirm the scaling behavior and offer a comprehensive explanation of the cyclic trajectory dynamics.

Comment 7:

In section IV of the supplementary, the detuning of the drive is again discussed as a tuned parameter but here it is in relation to the Trotterization technique where there is no drive and I don't understand what the meaning of detuning has with respect to a drive with no amplitude that does not play a role in the process. Please explain this in the text.

Response:

We thank the Reviewer for pointing out this and apologize for the confusion. As noted in our response to comment 4, the Trotterization approach in our experiment does not need to apply the detuned drive, so we set $\Omega_d = 0$ and apply a virtual phase rotation after each Kerr evolution step to compensate for Kerr-induced phase accumulation. To avoid any confusion and clarify this point, we have removed the description of frequency detuning and changed to virtual phases in the revised manuscript.

Comment 8:

In the abstract, the first line states, "Quantum squeezed states, with reduced quantum noise...". The word "reduced" here does not seem accurate and "biased noise" seems to be a more appropriate term as the noise will be reduced in one quadrature but enhanced in another for a squeezed state.

Response:

We thank the Reviewer for catching this inaccuracy. As suggested, we have revised the word “reduced” to “biased” in the abstract of the revised manuscript.

Comment 9:

Excessively self-congratulatory adjectives like “ingenious” (near the end of the introduction section) is not appropriate for a scientific publication.

Response:

We have removed these adjectives in the revised manuscript following the Reviewer’s suggestion.

Comment 10:

In the concluding paragraph, the authors made a strong claim without evidence that “the squeezing degree can be further enhanced by engineering a weaker Kerr nonlinearity and a larger displacement amplitude”. Weaker single-photon nonlinearity but a larger displacement does not necessarily mean weaker spurious higher-order terms. Furthermore, as the displacement amplitude increases and the average photon number becomes larger, would the increased photon loss rate cause problems for this scheme? The authors must provide much more in-depth analysis or even experimental tests to back this claim up. Otherwise, this claim should be removed.

Response:

We appreciate the Reviewer’s valuable comment and apologize for this confusion. In response, we have removed this statement from the main text and discussed in more detail in the revised Supplementary Material. To further enhance the squeezing degree in our experiment, the primary limitation is the maximum achievable displacement amplitude. The amplitude is constrained by the electronic hardware and the breakdown of the dispersive approximation at high photon numbers, which would induce parasitic qubit excitations during the evolution. The intrinsic Kerr nonlinearity of the cavity would also restrict the achievable photon number and introduce phase-space twisting.

To support the statement that a weaker Kerr nonlinearity could enable higher squeezing, we have performed numerical simulations, and the results are presented in Section VII and Fig. S13 in the revised Supplementary Material. The simulation results indicate that employing a weaker Kerr nonlinearity $K/2\pi = 1$ kHz would yield a state fidelity exceeding 0.99 with a squeezing parameter $|\xi| \approx 1.5$. In addition, we have discussed the limitations of the Trotterized squeezing approach in our experiment, and performed numerical simulations

incorporating the single-photon loss errors, dephasing errors, and higher-order Kerr nonlinearities to evaluate their influences on the state fidelity and squeezing parameter of the generated nonclassical states. A more detailed error analysis is also provided in Section VII of the revised Supplementary Material.

Comment 11:

Overall, the experiments performed and the approach to squeezed state generation are very nice and innovative and would find useful applications in the field. However, the presentation of the results, as written, needs some work. At the moment I cannot recommend publication but if the above-mentioned problems are fixed, I would recommend publication.

Response:

We greatly appreciate the Reviewer for their positive assessment of our work as “*very nice and innovative*”. We have thoroughly addressed all the comments raised by the Reviewer in the revised manuscript and believe that the clarifications and additions made have significantly improved the presentation. We hope the revised version now meets the high standards required for publication in *Nature Communications*.

Response to Reviewer #1 & # 4's Comments

Comment 1:

We appreciate the thorough revisions made the authors and the detailed responses to our previous questions. Overall, we believe that the manuscript is significantly more compelling compared to the earlier version. In particular, the new experimental results on amplitude estimation are very useful in supporting the claims of utility for the states they created despite the imperfections/deviations from ideal squeezed states. With these revisions, we believe that the manuscript meets the standards of Nature Communications and recommend publication after addressing a few additional questions listed below.

Response:

We thank the Reviewer for their positive assessment of our revised manuscript and for acknowledging that the new experimental results strengthen the metrological utility of the generated squeezed states. We are pleased that they find our manuscript now meets the standards of *Nature Communications* and recommend its publication. Below, we provide point-by-point responses to their additional questions.

Comment 2:

Why is Allan deviation used? I would think that the variance of the amplitude estimation can be directly inferred from the posterior distribution from the Bayesian method. Is the Allan deviation a more sensitive or robust measure? Could they directly provide the variances as well?

Response:

We thank the Reviewer for raising this point. In quantum sensing experiments, the Allan deviation serves as a well-established metric for characterizing measurement precision, particularly useful for identifying and quantifying different types of noise. While the Bayesian posterior distribution indeed allows for the direct estimation of the variance for a fixed set of measurements, the Allan deviation is a more sensitive measure for characterizing low-frequency correlated noise than conventional variance, which quantifies only Gaussian white noise across the entire dataset and fails to capture temporal noise correlations.

In response to the Reviewer's suggestion, we have now included the standard deviation (square root of the variance) extracted from the Bayesian posterior distributions as a function of the number of measurements in the following Figure 1 [see Supplementary Figure 13].

Figure 1: Extracted standard deviation (triangles) from the Bayesian posterior distributions for estimating the displacement amplitude using squeezed states generated with different Trotter steps (red: $M=1$, green: $M=13$) versus the number of independent measurements N_{meas} . Dashed lines represent the theoretical Cramér-Rao bound.

Comment 3:

Some treatments of the data are not presented, which are important for reproducibility. For example, the maximum operational point α_{opt} (this deviates from zero?); the fitting the Authors used to get a smooth p_g for the calculation of Fisher information; Fisher vs amplitude. A few exemplary plots would be nice.

Response:

We appreciate the Reviewer for this valuable suggestion aimed at improving reproducibility. To address this, we have added detailed descriptions and exemplary plots in Supplementary Note 5-A of the revised manuscript.

As shown in Figure 2 [see Supplementary Figure 12], we now include representative results for the measured qubit ground state population $P_g(\alpha)$ and the corresponding extracted classical Fisher information $I_F(\alpha)$ as a function of the displacement amplitude α (for $M=6$). The $P_g(\alpha)$ data are fitted using $P_g(\alpha) = A(\exp(-2e^{2|B|}\alpha^2)) + D$, which are used for calculating the Fisher information $I_F(\alpha)$ to extract the optimal operating point $\alpha_{\text{opt}} = \arg \max_{\alpha} I_F(\alpha)$ as clearly indicated in the figure. Notably, α_{opt} deviates from zero, indicating the displacement required to maximize the metrological sensitivity. Additionally, we provide a plot of the experimentally determined α_{opt} as a function of the number of Trotter steps M .

Figure 2: Extracting the optimal displacement amplitude for quantum sensing using the generated squeezed states. (a) Measured qubit ground state populations (blue symbols), corresponding fit (blue line), and extracted Fisher information (yellow line) as a function of the displacement amplitude α with $M=6$. The optimal operating point with maximum FI is indicated. (b) Extracted optimal displacements as a function of the number of Trotter steps M .

Comment 4:

The $p_g(\alpha)$ used Eq. S18 is theoretical or experimental fit?

Response:

The $P_g(\alpha)$ used in Eq. S18 refers to the experimental fit using the expression $P_g(\alpha) = A \left(\exp(-2e^{2|B|\alpha^2}) \right) + D$, where fitting parameters A and D account for experimental imperfections, and B corresponds to the extracted squeezing parameter (see Supplementary Note 5-A).

Comment 5:

In SM section V. A. Might be typo. Should the p_g in $p_g(\alpha) = 0.5(\pi/2 \dots)$ be p_e ?

Response:

We thank the Reviewer for their careful reading. This expression is indeed correct as written. In our displacement sensing experiment, we employ the parity measurement through a sequence involving a controlled-phase gate $C_\pi = e^{i\pi|e\rangle\langle e|}$ between $\hat{X}_{\pi/2}$ and $-\hat{X}_{\pi/2}$ qubit rotations. This sequence transforms the initial joint cavity-qubit state $\sum_n c_n |n\rangle |g\rangle$ into a final state $\sum_n [c_n |n\rangle (|g\rangle + i|e\rangle) - ie^{i\pi} c_n |n\rangle (i|g\rangle + |e\rangle)]/2$, from which the qubit ground state population is derived as $P_g = \sum_n \frac{1+e^{i\pi}}{2} |c_n|^2$. Since the Wigner function of the initial cavity state can be calculated as $W = \frac{2}{\pi} \sum_n |c_n|^2 e^{i\pi n}$, we thus derive $P_g = \frac{1}{2} \left[\frac{\pi}{2} W + 1 \right]$. Therefore, the expression presented in the manuscript is accurate.

Response to Reviewer #3 & # 5's Comments

Comment 1:

The additions made to the text were very good overall and the manuscript has really been elevated from the first draft. The addition of an intuitive picture to the results was very helpful in understanding the experiment. The addition of the metrology experiment gave much better motivation for the experimental technique and put the claims on much more solid footing. The derivation of the power law in the supplementary was very useful. A lot of wording and confusing explanations were cleared up, making the paper much more cohesive and understandable.

There are still three points we think should be addressed:

Response:

We sincerely thank the Reviewer for their positive assessment of our revised manuscript and for recognizing the added intuitive explanations, the new metrology experiments, and the improved theoretical derivations, all of which have significantly strengthened our work. We have now carefully addressed the three remaining points below.

Comment 2:

As we noted earlier, the Trotterization approach seemed easier to implement, achieved better results faster and was more intuitive to understand, while the driven approach seemed like more of a side-show and has no real advantage. We were therefore confused about the strong emphasis on the driven approach in the writing. This does not impact the overall significance/correctness of the work, but it just seems an awkward way to present the results. The authors added some sentences in the revision to segue these results a bit nicer, but didn't really justify this approach of presentation. If there is some distinct advantage to the driven approach, it should be made more clear.

Response:

We appreciate the Reviewer's insightful suggestion. In response, we have reorganized the presentation to better justify the driven approach and highlight the Trotterization method.

The driven approach serves as the conceptual foundation, demonstrating the fundamental cyclic squeezing dynamics of a Kerr nonlinear oscillator and offering potential applicability and flexibility in engineering complex squeezing evolution. Meanwhile, the Trotterization protocol builds upon this foundation to achieve enhanced squeezing performance with

experimental simplicity for generating large squeezed vacuum states.

To better contextualize this in the manuscript, we have added a schematic diagram (Fig. 1d) to intuitively illustrate and highlight the Trotterized squeezing technique. In addition, we have removed the original Fig. 3 and corresponding discussions to the Supplementary to avoid overemphasis on the driven squeezing approach, and added a sentence: "*In addition, the driven squeezing approach may offer potential applicability and flexibility in engineering complex squeezing evolution. For example, we employ this approach to generate squeezed multiphoton Fock states $S(\xi)|N\rangle$ with N up to 6 (see Supplementary Figure 7).*" in the last paragraph on page 3 in the main text of the revised manuscript.

These revisions better justify our presentation while highlighting the Trotterization squeezing technique as suggested by the Reviewer.

Comment 3:

There still seems to be insufficient evidence to support the claim that the squeezing degree can be increased with a larger displacement amplitude and weaker Kerr. The authors added an explanation for this statement with a simulation showing that with a larger displacement and smaller Kerr, the simulated fidelity is greater than 99%. However, it is stated that a predominant source of errors is the higher order Kerr terms such as 6th and 8th order terms. It would seem that these higher order terms scale such that they would have an even larger contribution as photon numbers get larger, making them a dominant error pathway in this discussion, so using a simulation with just the 4th order Kerr terms and single photon loss seems to be neglecting a very large error avenue for the statement that is trying to be made here. Either the higher order terms should be taken into account and shown concretely to still allow for higher fidelities, or this statement should be taken out of the text.

Response:

We thank the Reviewer for raising this point. We agree that higher-order Kerr nonlinearities (e.g., 6th- and 8th-order terms) would distort the squeezing evolution when using large displacements, and should be included in the simulation to avoid overestimating the achievable squeezing performance. In our simulation, we directly compare the squeezing fidelity considering only the Kerr term $\frac{K}{2}a^\dagger a^2$ ($K/2\pi = 1\text{kHz}$) and including the higher-order Kerr terms $\frac{K_2}{6}a^\dagger a^3$ and $\frac{K_3}{24}a^\dagger a^4$ ($K_2/2\pi = 5\text{Hz}$, $K_3/2\pi = 0.025\text{Hz}$), yielding a fidelity of 0.99 and 0.92, respectively, after five Trotterization steps. The results indicate that higher-order Kerr nonlinearities indeed introduce a certain degree of fidelity loss. However, it is challenging to accurately characterize these small strengths of high-order Kerr

nonlinearities in our experiment. Therefore, in accordance with the Reviewer's suggestion, we have removed the corresponding description from the Supplementary Information to avoid potential overclaim.

Comment 4:

The additional experiments and writeup on the newly added Quantum metrology section are a great addition to the paper and make the message much more convincing. The only point to add here is an explanation for the data point in Fig. 5b) for the $m=13$ point at the highest N_{meas} would be nice, it is odd that this point is so far off the trend line and outside of any statistical fluctuation based on the size of the error bars.

Response:

We appreciate the Reviewer for recognizing the value of the newly added metrology experiments. The increase observed in the last data point of the Allan deviation for $M = 13$ indicates the presence of low-frequency correlated noise in the system, as Allan deviation is a well-established sensitive metric for identifying and quantifying such low-frequency fluctuations. This deviation from the trend line may be primarily attributed to low-frequency displacement drifts in the experimental apparatus. As suggested, we have added a brief note: "The increase of the last point for $M = 13$ may primarily be attributed to low-frequency displacement drifts in the experimental apparatus." in the caption of Fig. 4b in the revised manuscript.

In this article, the authors presented a scheme to squeeze vacuum and Fock states in a superconducting microwave cavity. They use the weak Kerr non-linearity of the cavity with an off-resonant drive. The resulting action resembles a squeezing operation, whose effective strength is enhanced during the evolution by the strong displacement in phase space. To mitigate the effect of the unwanted term in the Hamiltonian (second line in Eq.2), the authors employed Trotterization method. The experimental results in the paper show states that resemble highly squeezed vacuum and Fock states.

The positive aspects of this papers are:

1. The method does not use the qubit in the evolution, which makes it not prone to qubit decoherence.
2. The experimental technique is simple and only uses natively available gates and dynamics. Thus, it is easily implemented on typical bosonic cQED setups.

However, despite these interesting aspects, we have some strong concerns about the results:

1. The authors only presented the degree of squeezing as a figure of merit. To us, this does not capture the full picture. The states visually look at squeezed states but have many artefacts, as the Kerr dynamics is not the same as squeezing. So to claim that strongly squeezed states are produced seems to oversimplify what is going on. What about the overall quality of the generated states, i.e., the fidelity of the generated state to the target squeezed states?
2. The description of the analysis is also quite vague. They mention that they perform fit to extract the squeezing level. What about the fitting errors? This is particularly a good thing to show as one can achieve high squeezing level but the state smears out in phase space (like in Fig. 4d $M=13$). What about the anti-squeezing level on these states? Do they make sense? It is not complete to put maximum squeezing 14.6 dB in the abstract without also presenting how good the quality of the state is.
3. The authors mentioned that at the end they displaces back the compressed coherent state back to origin and perform a virtual phase rotation to eliminate rotation angle. What is then the role of the cycle, as one can stop at any point in Fig. 2b and performed the mentioned corrections?
4. Ideally, squeezed vacuum does not have Wigner negativity, it is simply a vacuum state with positive Wigner function squeezed in one quadrature. The Wigner logarithmic negativity that the authors computed are caused by imperfections and does not show the main non-classical feature of the squeezed state.
5. From what is presented in the manuscript, there is little evidence that the states created as close to ideal squeezed states. Given this, are the reported states actually useful for anything that a typical squeezed vacuum is suited for? The last figure computes the FI, but it is not in the context of any real tasks that uses the state (such as metrology). It does not seem compelling or obvious to us that such states are useful given their pronounced artefacts and deviations from ideal squeezing. To show us, the authors would need to

perform a real experiment using this state, which I think would require a rather major revision.

Overall, we found that the manuscript falls short of the standards of Nature Communications and do not recommend publication.

Review report for resubmitted manuscript

Quantum squeezing amplification with a weak Kerr nonlinear oscillator

We appreciate the thorough revisions made the authors and the detailed responses to our previous questions. Overall, we believe that the manuscript is significantly more compelling compared to the earlier version. In particular, the new experimental results on amplitude estimation are very useful in supporting the claims of utility for the states they created despite the imperfections/deviations from ideal squeezed states. With these revisions, we believe that the manuscript meets the standards of Nature Communications and recommend publication after addressing a few additional questions listed below:

1. Why is Allan deviation used? I would think that the variance of the amplitude estimation can be directly inferred from the posterior distribution from the Bayesian method. Is the Allan deviation a more sensitive or robust measure? Could they directly provide the variances as well?
2. Some treatments of the data are not presented, which are important for reproducibility. For example, the maximum operational point α_{opt} (this deviates from zero?); the fitting the Authors used to get a smooth p_g for the calculation of Fisher information; Fisher vs amplitude. A few exemplary plots would be nice.
3. The $p_g(\alpha)$ used Eq. S18 is theoretical or experimental fit?
4. In SM section V. A. Might be typo. Should the p_g in $p_g(\alpha)=0.5(\pi/2\dots)$ be p_e ?